# Enhanced Degradation of Rhodamine B by Metallic Organic Frameworks Based on NH_2_-MIL-125(Ti) under Visible Light

**DOI:** 10.3390/ma14247741

**Published:** 2021-12-15

**Authors:** Hong-Tham Nguyen Thi, Kim-Ngan Tran Thi, Ngoc Bich Hoang, Bich Thuy Tran, Trung Sy Do, Chi Sy Phung, Kim-Oanh Nguyen Thi

**Affiliations:** 1Institute of Environmental Sciences, Nguyen Tat Thanh University, Ho Chi Minh City 70000, Vietnam; nguyenhongtham0521@gmail.com (H.-T.N.T.); nganttk@ntt.edu.vn (K.-N.T.T.); bichhn@ntt.edu.vn (N.B.H.); pcsy@ntt.edu.vn (C.S.P.); 2Vo Giu High School, Hoai An District, Binh Dinh Province 55000, Vietnam; 3Faculty of Environmental and Food Engineering, Nguyen Tat Thanh University, Ho Chi Minh City 70000, Vietnam; 4Institute of Environmental Science, Engineering and Management, Industrial University of Ho Chi Minh City, Ho Chi Minh City 70000, Vietnam; tranbichthuy@iuh.edu.vn; 5Institute of Chemistry, Vietnam Academy of Science and Technology, Hanoi City 10000, Vietnam; dosyvhh@gmail.com

**Keywords:** metal–organic framework, RhB, bimetallic, degradation, kinetic

## Abstract

Samples of the bimetallic-based NH_2_-MIL-125(Ti) at a ratio of M^n+^/Ti^4+^ is 0.15 (M^n+^: Ni^2+^, Co^2+^ and Fe^3+^) were first synthesized using the solvothermal method. Their fundamental properties were analyzed by X-ray diffraction (XRD), Fourier transform infrared spectroscopy (FTIR), Raman spectra, scanning electron microscopy (SEM), N_2_ adsorption–desorption measurements, and UV–Vis diffuse reflectance spectroscopy (UV-Vis DRS). The as-acquired materials were used as high-efficiency heterogeneous photocatalysts to remove Rhodamine B (RhB) dye under visible light. The results verified that 82.4% of the RhB (3 × 10^−5^ M) was degraded within 120 min by 15% Fe/Ti−MOFs. Furthermore, in the purpose of degrading Rhodamine B (RhB), the rate constant for the 15% Fe/Ti-MOFs was found to be 2.6 times as fast as that of NH_2_-MIL-125(Ti). Moreover, the 15% Fe/Ti-MOFs photocatalysts remained stable after three consecutive cycles. The trapping test demonstrated that the major active species in the degradation of the RhB process were hydroxyl radicals (HO^∙^) and holes (h^+^).

## 1. Introduction

Rapid industrial expansion in recent decades has led to a markedly increased discharge of highly hazardous substances into water systems [1]. The presence of organic matter, especially colorants, hair dye [2], leather and paper industries [3], and luminescent solar concentrator (LSC) technologies [4] in water streams is usually undesirable, even in small concentrations, because it can obstruct the entry of light and oxygen to the water body, thus reducing photosynthetic and aquatic creature activity [5].

Among industries, the textile sector can generate a substantial amount of unstable, difficult-to-degrade colorants, thus highlighting the importance of techniques for the removal of existing chemical contaminants from wastewater to reduce the negative effects of sewage disposal.

Despite rapid development regarding a wide array of semiconductor photocatalysts for organic pollutant degradation, the quantum yield and solar energy conversion efficiency of photocatalysts remain poor, limiting their practical usage in various field [6,7,8,9]. Therefore, successful introduction of new high-efficiency photocatalysts for pollutant degradation remains a significant challenge.

Metal–organic frameworks (MOFs) are a type of newly developed porous coordination polymer composed of metal ions/clusters linked by organic ligands [10]. When exposed to light, MOFs have recently been discovered to operate as semiconductors, making them suitable photocatalysts for a variety of reactions [11,12]. NH_2_-MIL-125(Ti), which has the chemical formula Ti_8_O_8_(OH)_4_(BDC-NH_2_)_6_, contains an amine functionalized with the Ti-based MOFs (MIL-125(Ti) [13]. Previous studies have shown that NH_2_-MIL-125(Ti) has the capacity to photocatalytically degrade colorants under visible light with excellent performance [13,14,15,16,17]. The red adsorbent converts the Ligand-to-metal cluster charge transfer (LMCCT) absorption band to a visible region for NH_2_-MIL-125(Ti) by adding the amino autochrome group to the terephthalate-binding precursor [14,18]. This group, along with other linker substitution groups, alters the overall electronic structure of the MIL-125 framework but has negligible effect on the uncoupled orbitals on the Ti-oxo node cluster, limiting MOFs optical exploitation.

The substitution of different metals into the Ti-oxo cluster of NH_2_-MIL-125(Ti) as a method to transform empty orbitals has received much scientific attention [10,19,20]. In a previous study, Ti/Zr-MOFs were synthesized by partially replacing Ti with Zr at a ratio of Zr^4+^:Ti^4+^ of 0.15, and the as-synthesized materials exhibited the highest degradation of RhB [21]. Hong Liu et al. [10] also successfully synthesized Cu-doped NH_2_-MIL-125(Ti), which showed enhanced photocatalytic performance for degrading methyl orange (MO) and phenol. The method described above showed that doping transition metals into the structure of MOFs could result in a range of hybrid MOFs with improved photocatalytic activity. However, recent studies on the effect of different transition metal clusters on photocatalytic efficiency are rare.

In this communication, we attempted to synthesize NH_2_-MIL-125(Ti) and modify the parent material by transition metals at a ratio of M/Ti is 0.15 (M: Ni, Co, and Fe) via the solvothermal method. A series of 15% M/Ti-MOFs crystals were obtained and characterized using various methods, including XRD, SEM, FTIR, Raman, UV-Vis Diffuse reflectance spectroscopy (UV-Vis DRS) and BET. The photocatalytic activity of the obtained MOFs was studied by degrading Rhodamine B under visible light.

## 2. Materials and Methods

### 2.1. Preparation of Materials

#### 2.1.1. Materials

Titanium (IV) isopropoxide (Ti(OiPr)_4_) and 2-amino terephthalic acid (H_2_NC_6_H_3_-1,4-(CO_2_H)_2_) were purchased from Sigma-Aldrich Co. (St. Louis, MO, USA). Iron(III) chloride hexahydrate (FeCl_3_·6H _2_O), nickel chloride hexahydrate (NiCl_2_·6H_2_O), and cobalt(II) chloride hexahydrate (CoCl_2_·6H_2_O) were obtained from Fisher Scientific (Fair Long, NJ, USA). N-dimethylformamide (DMF, 99.5%) and methanol (CH_3_OH) were bought from Xilong Chemical Co., Ltd. (Guangdong, China).

#### 2.1.2. Preparation of NH_2_-MIL-125(Ti) and 15% M/Ti-MOFs (M: Ni, Co, Fe)

NH_2_-MIL-125(Ti) photocatalyst was synthesized using the solvothermal method based on the previously described method [19]. Briefly, after dissolving 0.504 g 2-amino-1,4-benzene dicarboxylate (H_2_NC_6_H_3_-1,4-(CO_2_H)_2_, Sigma-Aldrich Co., St. Louis, MO, USA) in 12.1 mL N, N-dimethylformamide (DMF), 1.4 mL methanol was added to the mixture and magnetically agitated for 30 min at room temperature. After that, 0.26 mL titanium isopropoxide (Ti(OiPr)_4_) was added, followed by stirring until the mixture became homogeneous and then by addition of 0.02 g CTAB surfactant (Sigma-Aldrich Co., St. Louis, MO, USA). The reaction mixture was then transferred to a 100 mL Teflon-lined autoclave (TEFIC Biotech Co., Xi’an, China) and planted in a closed convection heat device at 150 °C for 24 h. To eliminate BDC-NH_2_ from the pores, the product was chilled and then re-dispersed in DMF overnight at 80 °C. The resulting products were centrifuged, rinsed several times by DMF and methanol solvent, and then vacuum-dried overnight at 150 °C. 

M-doped NH_2_-MIL-125(Ti) was synthesized through the above similar process, except that pure Ti[OCH(CH_3_)_2_]_4_ was replaced with a ratio M^n+^/Ti^4+^ is 0.15. Herein, the salts used as sources to provide the transition metal include NiCl_2_·6H_2_O, CoCl_2_·6H_2_O, and FeCl_3_·6H_2_O. The products were denoted as 15% M/Ti-MOFs.

### 2.2. Instrumentation

The obtained catalysts were characterized by powder X-ray diffraction (PXRD) technique, performed by a 600 XRD diffractometer (Shimadzu, Kyoto, Japan) with Cu Kα radiation (λ = 1.5418 Å). For Raman spectroscopy analysis, a HORIBA Jobin Yvon spectrometer (Horiba Scientific, Kyoto, Japan) was operated in the wavenumber range of 100–1000 cm^−1^ (633 nm of the laser beam). The scanning electron microscopy (SEM) was conducted on a JEOL JSM 7401F (Peabody, MA, USA). Fourier Transform Infrared spectroscopy (FTIR) was recorded on a Jasco-4700 FTIR Spectrometric Analyzer (Jasco International Co., Ltd., Tokyo, Japan). N_2_ absorption/desorption was analyzed using a Micromeritics Tristar 3000 (Micromeritics instrument corporation, Norcross, GA, USA) at 77 K. The samples were kept at 150 °C for 12 h to degas. The optical properties of the as-acquired materials were determined by UV–Vis DRS (Ultraviolet-visible diffuse reflectance spectroscopy) (UV-2450, Shimadzu, Kyoto, Japan). 

### 2.3. Photocatlytic Test

All synthesized MOF catalysts underwent the degradation process under visible light against RhB dye (Sigma-Aldrich Co., St. Louis, MO, USA) to determine their photocatalytic performance. The mixture included a 5 mg catalyst, 100 mL RhB (3 × 10^−5^ M), and the aid of H_2_O_2_ (1 mL, (H_2_O_2_) = 2 mM). The mixture was contained in a 250 mL double-layer interlayer glass beaker. In this study, a LED lamp (40 W, Cree., Inc., Durham, NC, USA) acted as the visible light source. The suspensions were magnetically stirred for 1 h without light to balance adsorption and desorption. A volume of 4 mL solution was taken during the photodegradation process and centrifuged at 6000 rpm at regular intervals for 10 min to separate the solid material. A UV-Vis spectrophotometer (Agilent Cary 60, Agilent Technologies, Santa Clara, CA, USA) was used to measure the dye concentration and absorption at the wavelength of 554 nm.

### 2.4. Trapping Test

The RhB photodegradation mechanism of 15% Fe/Ti−MOFs was carried out using several scavengers. Herein, p-benzoquinone (BQ, Sigma-Aldrich Co., St. Louis, MO, USA), tert-butanol (TBA, Sigma-Aldrich Co., St. Louis, MO, USA), and EDTA-2Na (Sigma-Aldrich Co., St. Louis, MO, USA) were used to inhibit O_2_^−^, HO^−^, and h^+^ scavengers, respectively. TBA and EDTA concentrations were 1 mM, while the BQ concentration was 1 μM.

## 3. Results and Discussion

### 3.1. Characterization

The XRD pattern of NH_2_-MIL-125(Ti) (between the 2θ range of 5–40°) matches perfectly with the simulated XRD pattern, indicating that the synthesized MOFs had high crystallinity with the featured peaks, including those at 6.8°, 9.7°, 11.6°, 15.2°, and 19.5° [7]. The XRD patterns for the bimetallic doped NH_2_-MIL-125(Ti) samples show the signature diffraction peaks corresponding to NH_2_-MIL-125(Ti), thus suggesting that the structures of the MOFs have entirely stabilized. However, with the doping of transition metals, the diffraction peaks are somewhat shifted to lower angles, as illustrated in Figure 1B.

The Raman spectra of NH_2_-MIL-125(Ti) and 15% M/Ti-MOFs contribute to further corroboration of the framework structure in MOFs (Figure 2A). The characteristic peaks in all of these spectra are similar. Relatively strong bands of NH_2_-MIL-125(Ti) were observed at 546, 640, 1145, and 1620 cm^−1^ [22]. The band characterizing the N−H bond of the organic linker was located at 1620 cm^−1^ [23]. The appearance of a band of symmetrical bending and elongation at 546 cm^−1^ implies the presence of the Ti-O-Ti-O octahedral ring species [10,18].

The FTIR spectra of NH_2_-MIL-125(Ti) and the modified 15% M/Ti-MOFs samples were detailed in Figure 2B. The drawbacks in carboxylate-based materials are frequently linked to partial structural hydrolysis, resulting in the formation of hydroxo species and in turn causing extra peaks in the 3700–3500 cm^−1^ range and broad-band H−bond interaction in the range of 3600–2500 cm^−1^ [24]. However, the peak intensity for Co- and Fe-doped MOFs decreased compared to the original MOF. One possibility to explain this trend is related to the structural inhomogeneity of the mixed-metal MOFs. Although XRD analysis (Figure 1) and FTIR spectroscopy (Figure 2B) showed no signs of extended impurity phases, a locally restricted uneven metal apportionment cannot be excluded. An alternative explanation indicates a more complex interplay between the different metal-oxo clusters that coordinate to the same ligand.

All samples showed a doublet band at from 500 to 800 cm^−1^ which could be attributed to the Ti–O vibration and an oxo-TiO cluster of the metal–organic framework [25]. Besides, the same trend was observed for the four materials that showed the feature vibration for NH_2_-BDC ligand. Essentially, the asymmetric stretching vibrations of −COO^−^ at 1537–1599 cm^−1^, the symmetric stretching of −COO^−^ at 1383 cm^−1^ and a peak at 1256 cm^−1^ indexed to the expanded vibrations of the C−N bond from the aromatic amine were observed in spectra of all of the samples, indicating the successful coordination NH_2_-BDC ligands into the framework [26,27,28]. These results further proved the presence of the NH_2_-MIL-125(Ti) framework in the bimetallic materials. 

The surface morphologies of NH_2_-MIL-125(Ti) and 15% M/Ti-MOFs (M: Ni, Co, and Fe) were studied through SEM images (Figure 3). The collected samples were composed of several well-crystallized block-like particles of varied forms, as seen in the low-magnification SEM images (Figure 3). The NH_2_-MIL-125(Ti) catalyst has thin and disc-shaped particles with an average size of 0.7 µm, similar to that of the previously studied catalyst [19,20]. The grain size increased clearly when NH_2_-MIL-125(Ti) was modified with the transition metal ions.

The BET surface area and pore structure of the as-acquired samples were explored by the N_2_ adsorption/desorption isotherms, as displayed in Figure 4. These isotherms are type-I characteristics for a micropore structure without mesopores (Figure 4A). After encapsulation, the reduction of BET surface area (970.2 vs. 889.6, 640.7, and 554.9 m^2^ g^−1^) and pore volume values and (0.94 vs. 0.81, 0.47, and 0.52 cm^3^ g^−1^, Table 1) confirmed the location of the transition metals in the pores of NH_2_-MIL-125(Ti). From Figure 4B, it can be seen that due to the encapsulation of transition metals within the porosity, the pore size distribution of MOFs materials is shifted to smaller sizes. 

The optical properties of NH_2_-MIL-125(Ti) and 15% M/Ti-MOFs (M: Ni, Co, Fe) were examined via the UV-Vis DRS spectra. It was found that the bimetallic MOF samples synthesized with different metals presented absorption spectra similar to that of the parent NH_2_-MIL-125(Ti) (Figure 5A). The feature characterizations for the optical properties of the MOFs were identical to those previously reported, with two peaks with maxima at 217 and 370 nm, related with Ti−O oxo-cluster absorption and the ligand-to-core charge transfer (LCCT), respectively [29,30]. However, for the transition metal-assisted samples, there was a slight blue shift in the absorption edge. Despite this, all four samples exhibited good optical absorption in the visible light range, demonstrating that they could be useful photocatalysts. In addition, the experimental bandgap (calculated as the UV-Vis spectrum’s lowest energy inflection point) is predicted to be 2.56 eV, 2.53 eV, 2.52 eV, and 2.40 eV for NH_2_-MIL-125(Ti), 15% Ni/Ti−MOFs, 15% Co/Ti−MOFs, and 15% Fe/Ti−MOFs, respectively (Figure 5B, Table 2).

### 3.2. Photocatalytic Activity and Mechanism

To evaluate the photocatalytic performace of the MOFs samples, experiments on the degradation of RhB dye under visible light were carried out. Herein, the reaction system contained 5 mg catalyst, 100 mL RhB solution (3 ×10^−5^ M), and the aid of H_2_O_2_ 2 mM. The mean results calculated from triplicate photocatalytic tests are shown in Figure 6. In photocatalysis, the decomposition of organic molecules can be divided into three steps: pollutant adsorption; chemical reaction; and the desorption of water, carbon dioxide, and by-products [31]. Of these steps, pollutant adsorption is an important stage conducted in the dark phase. At this stage, the adsorption and desorption processes take place alternately leading to the change of C/Co in the course of a 1-hour reaction. As can be seen in Figure 6A, there was a little removal of RhB when using NH_2_-MIL-125(Ti), whereas all the samples of 15% M/Ti-MOFs showed high removal efficiency. To be specific, the removal of RhB owing to the degradation was enhanced from 52.1% (NH_2_-MIL-125(Ti)) to 55.6% (15% Ni/Ti-MOFs), 55.7% (15% Co/Ti-MOFs) and 82.4% (15% Fe/Ti-MOFs). Because of weak absorption wavelengths in the visible light range, the photocatalytic efficiency of NH_2_-MIL-125(Ti) is relatively low. Meanwhile, the hierarchical pore system formed in the 15% Fe/Ti-MOFs catalysts is responsible for its high degradation capability. Moreover, the doping with transition metal ions lowers the bandgap energy of the semiconductor material, preventing recombination of the e^−^ and h^+^ pairs and thus increasing their degradation performance. 

To further elaborate the reaction kinetics of the RhB degradation catalyzed by the MOFs photocatalysts, the pseudo-first-order model was calculated, as illustrated in Figure 6B. The rate value (k) of RhB degradation for these four samples were 15% Fe/Ti-MOFs (13.49.10^−3^ min^−1^) > 15% Co/Ti−MOFs (5.41.10^−3^ min^−1^) > 15% Ni/Ti-MOFs (5.35.10^−3^ min^−1^) > NH_2_-MIL-125(Ti) (5.2.10^−3^ min^−1^) (Table 2). According these results, 15% Fe/Ti-MOFs has the highest photocatalytic performance and rate kinetic in RhB degradation. Hence, this photocatalyst was used for the next measurements.

Figure 7 presents the UV–Vis absorption spectra of the RhB under the visible light for these four photocatalysts. Jiang et al. [32] and Zhang et al. [33] indicate that under photocatalytic circumstances, RhB dye can undergo small alterations such as de-ethylation, resulting in a change in the absorption peak, as illustrated in Figure 7. The RhB dye solution’s absorption peak was at 554 nm, and it gradually decreased due to dye degradation, eventually reaching its lowest value at 120 min. The photocatalyst cleavage of the aromatic ring of the dye molecules and the initiation of its degradation are thought to be the cause of the decrease in absorption peaks [33]. 

The effect of solution pH and initial concentration dye on the RhB degradation process was investigated, as shown in Figure 8A,B. The pH solution is an essential parameter, which affects the surface charge of the catalyst, the degree of ionization, and the speciation of the catalyst during degradation. The experiment was tested with an initial RhB concentration of 3.10^−5^ M, 5 mg 15% Fe/Ti−MOFs with pH ranging from 2.0 to 8.0 and the results are displayed in Figure 8A. After 120 min of irradiation, the RhB dye removal efficiency was 97.56% (pH 2), 97.76% (pH 4), 88.78% (pH 6), and 17.05% (pH 8). These results suggest that, in general, lower pH values are more favorable in removing RhB dye. At acidic pH values, protonation of the photocatalyst surface increases the electrostatic interaction with dye molecules, resulting in increased adsorption and, eventually, photodegradation [23,28]. Furthermore, in acidic media, the concentration of H^+^ ions results in a huge production of hydroxyl radicals, resulting in a high photodegradation extent of organic pollutants. Negatively charged sites, on the other hand, are numerous in alkaline media. This property leads to the electrostatic repulsion of dye molecules and poor dye breakdown. Moreover, the number of insoluble chemicals created increases during the photodegradation reaction. As a result, the intensity of the light transmitted is reduced, preventing the production of hydroxyl radicals and lowering photodegradation efficiency.

Figure 8B shows the effect of initial dye concentration on the photodegradation of RhB when using 15% Fe/Ti-MOFs. Notably, the highest and lowest RhB removal occurred at initial dye concentrations of 2.10^−5^ M and 4.10^−5^ M, which reached the efficiency of 87.66% and 72.81%, respectively. The removal effectiveness was also improved by increasing the contact duration between the dye molecules and the catalyst molecules. This could be explained by the greater density of holes, and hence the enhanced generation of H^+^ and OH^−^ ions with prolonged time at low initial concentrations [34]. Naturally, the high density of hydroxyl radicals and the easily excitation photocatalyst nanoparticles account for the improved photodegradation efficacy [35]. In contrast, the density of hydroxyl radicals is insufficient to destroy the dye molecules if the initial dye concentration continues to rise. As a result, photodegradation performance diminishes at high initial dye concentrations. The absorption of light by dye molecules, which hinders translocation, is another factor contributing to the decline in photodegradation efficiency [36].

Organic molecules, particularly those with double bonds, will be attacked by free radicals created during the breakdown in photochemical processes. These radicals largely determine the photocatalytic reaction process. To clarify the mechanism, EDTA, TBA and BQ were added to the reaction as electron capture agents for h^+^, HO and O_2_^−^ under fixed reaction conditions (RhB = 3.10^−5^ M, g 15% Fe/Ti-MOFs and H_2_O_2_ = 2.10^−3^ M). When electron capturers were added to the reaction system, the photocatalytic efficiency was dramatically lowered compared to the initial sample (Figure 9A). The greatest noticeable alteration was seen in the sample that had TBA and EDTA. This demonstrates that HO^−^ and h^+^ play a dominant role in the RhB degrading response of 15% Fe/Ti−MOFs. Hanna et al. [19] expected that the photo-excited electrons would similarly localize onto the Fe^3+^ sites upon the LMCCT state of NH_2_-MIL-125(Ti, Fe), yielding long-lived transient Fe^2+^ species. These species produce HO_2_/O_2_ ^−^, then the free radicals O_2_ ^−^ participate in the reaction to form HO radicals, which attack and discolor the dye molecules. Furthermore, h^+^ in the Highest Occupied Molecular Orbital (HOMO) region of the dye molecule and a strong oxidizing catalyst participate in the dye degradation and attack reaction [24] (Figure 9B).

The reusability of a catalyst is a significant aspect of its effectiveness in factual applications. Here, the reusability of 15% Fe/Ti-MOFs was repeated three times under the same conditions. After each activity assessment cycle, to maintain a 100 mL RhB solution at 3.10^−5^ M in the next experiment, the exact same volume and concentration were implemented in the reaction system. Figure 10A depicts the photodegradation of RhB after three runs. The removal of RhB decreased from 76.88% to 71.19%, indicating that the catalyst can be used repeatedly with no discernible variation in photocatalytic activity. The XRD pattern (Figure 10B) of the before and after degradation samples (used three timed) suggests good stability of the 15% Fe/Ti-MOFs after three runs. 

## 4. Conclusions

This paper describes the manufacture of monodispersed NH_2_-MIL-125(Ti) and bimetallic M/Ti-MOFs with good crystallinity, utilizing a simple, safe, and low-cost process. This study found that 15% M/Ti-MOFs have a high photocatalytic activity for RhB decomposition when exposed to visible light. The doped transition metal had a substantial influence on the photocatalytic activity of 15% M/Ti-MOFs. Its superior activity was attributed to the synergistic effect of the small size of the iron ion and NH_2_-MIL-125(Ti). Specially, after 120 min of irradiation, the 15% Fe/Ti-MOFs catalyst had the highest activity, removing 82.4% of the RhB dye. Given the ease with which 15% Fe/Ti-MOFs can be prepared and their high stability, this appears to be a potential candidate as a photocatalyst and in other relevant fields requiring stable solutions.

## Figures and Tables

**Figure 1 materials-14-07741-f001:**
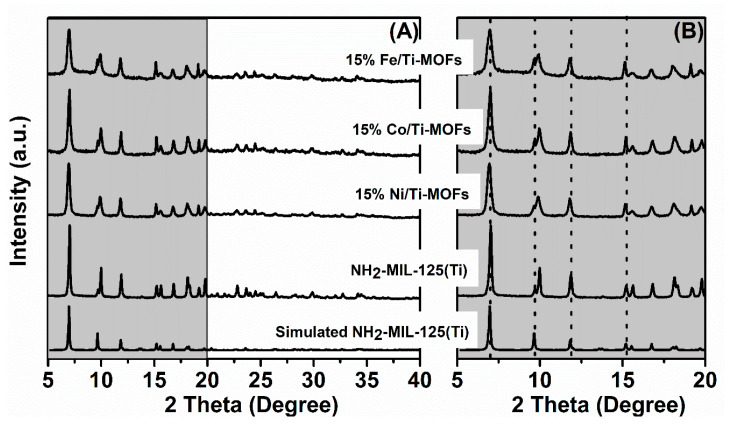
XRD parterns of NH_2_-MIL-125(Ti) and 15% M/Ti-MOFs (M: Ni, Co, Fe): (**A**): the 2θ range of 5–40° (**B**): the 2θ range of 5–20°.

**Figure 2 materials-14-07741-f002:**
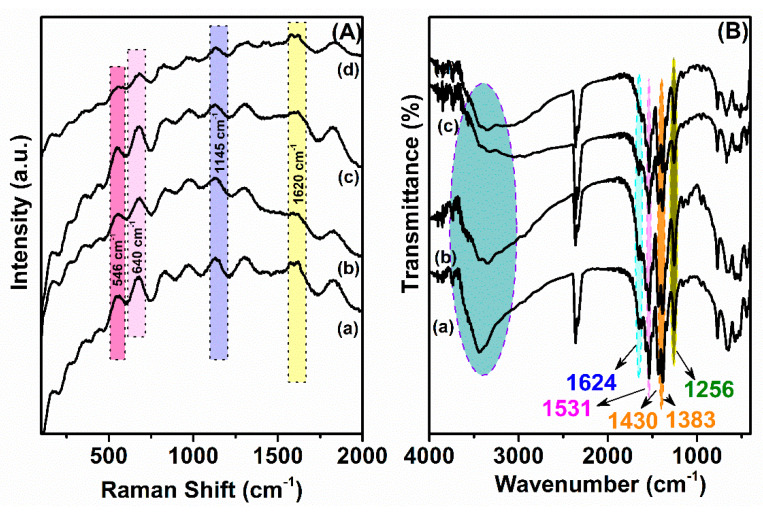
Raman spectra (**A**) and FT-IR spectra (**B**) of NH_2_-MIL-125(Ti) (a), 15% Ni/Ti-MOFs (b), 15% Co/Ti-MOFs (c), and 15% Fe/Ti-MOFs (d).

**Figure 3 materials-14-07741-f003:**
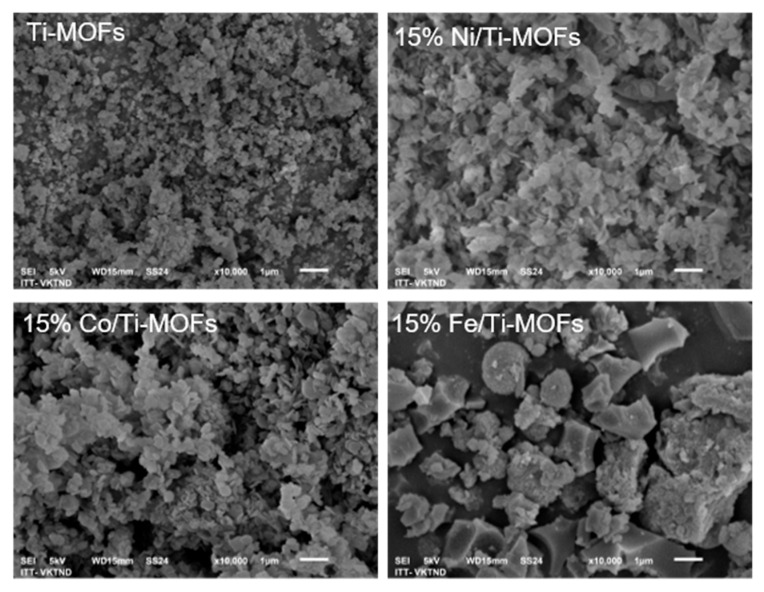
SEM image of NH_2_-MIL-125(Ti) and 15% M/Ti-MOFs (M: Ni, Co, Fe).

**Figure 4 materials-14-07741-f004:**
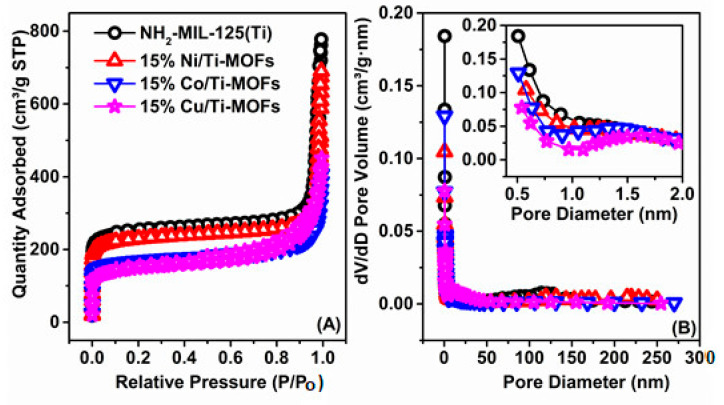
Adsorption–desorption isotherms (**A**) and pore size distribution (**B**) of NH_2_-MIL-125(Ti) and 15% M/Ti-MOFs (M: Ni, Co, Fe).

**Figure 5 materials-14-07741-f005:**
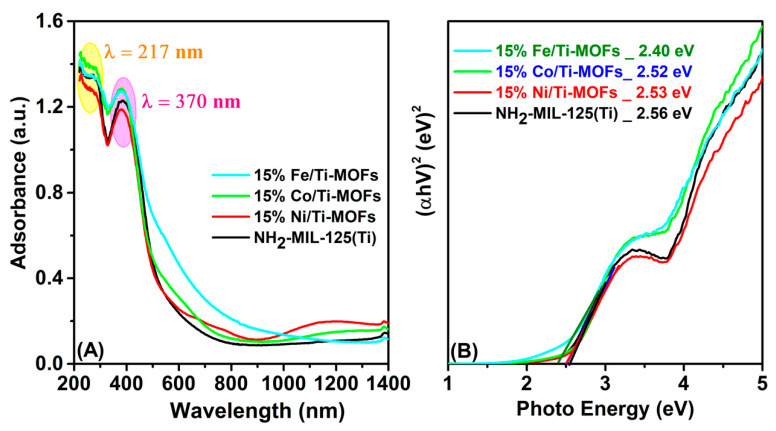
UV-Vis DRS (**A**) and Tauc plot (**B**) of NH_2_-MIL-125(Ti) and 15% M/Ti-MOFs (M: Ni, Co, Fe).

**Figure 6 materials-14-07741-f006:**
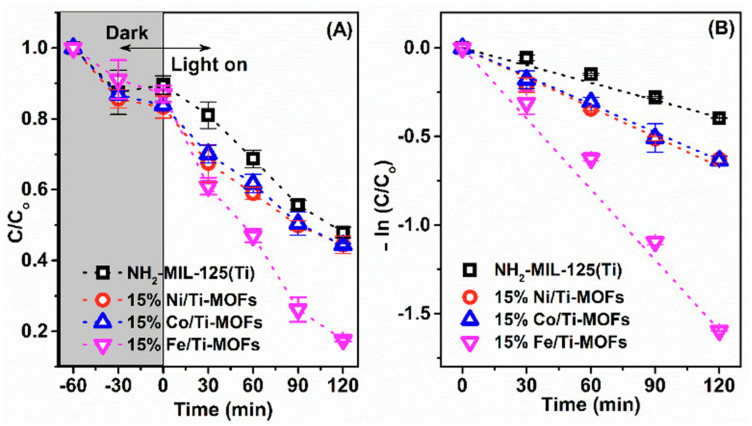
Removal efficiency of Rhodamine B using different photocatalysts under visible light irradiation with the aid of H_2_O_2_ (**A**), and pseudo-first-order kinetic of degradation process (**B**). Reaction conditions: Rhodamine B 3.10^−5^ mg/L, H_2_O_2_ 2 mM, catalyst 5 mg.

**Figure 7 materials-14-07741-f007:**
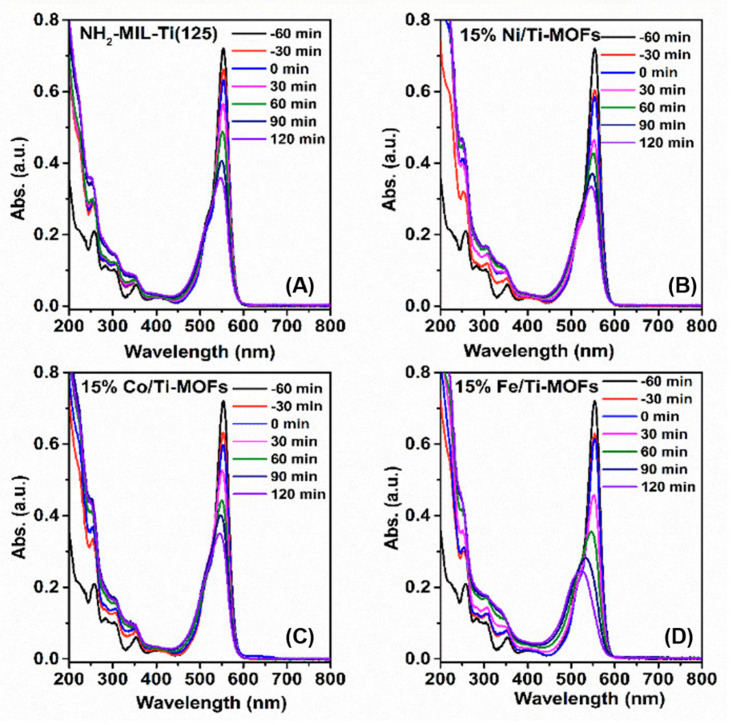
UV-Vis absorption spectra of RhB with respect to irradiation time over (**A**) NH_2_-MIL-125(Ti), (**B**) 15% Ni/Ti-MOFs, (**C**) 15% Co/Ti-MOFs, (**D**) 15%Fe/Ti-MOFs, respectively.

**Figure 8 materials-14-07741-f008:**
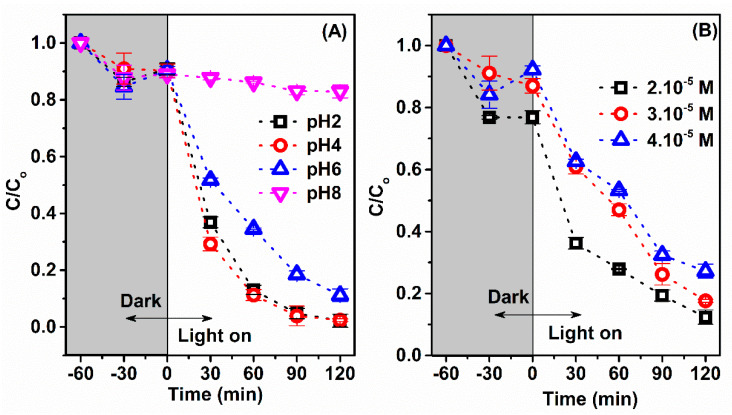
Effect of pH solution (**A**) and initial concentration dye (**B**) on Rhodamine removal efficiency over 15% Fe/Ti-MOFs.

**Figure 9 materials-14-07741-f009:**
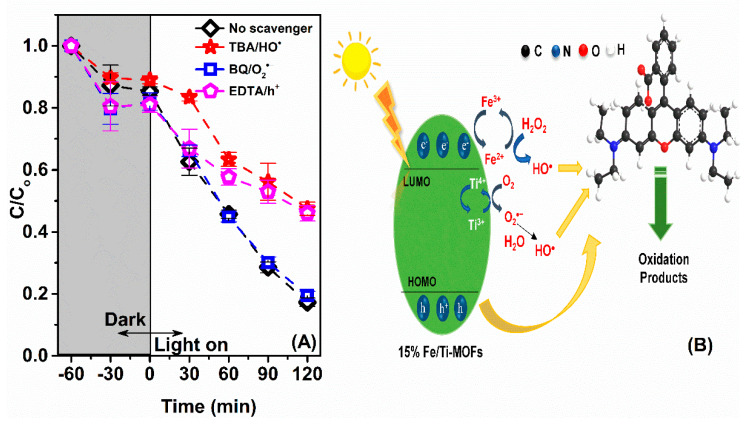
(**A**) Effect of radical scavengers; (**B**) Illustration of the proposed reaction mechanism on Rhodamine degradation over 15% Fe/Ti-MOFs. Reaction conditions: Rhodamine B 3.10^−5^ M, H_2_O_2_ 2 mM, 5 mg 15% Fe/Ti-MOFs catalyst.

**Figure 10 materials-14-07741-f010:**
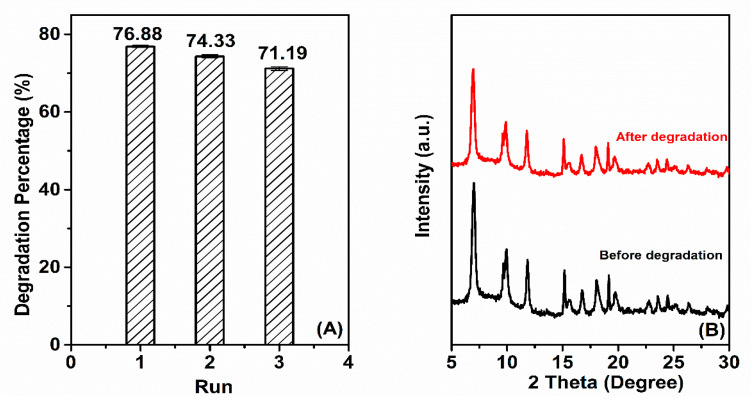
The reability test of the degradation of RhB over 15% Fe/Ti-MOFs (**A**), and the XRD pattern of 15% Fe/Ti-MOFs before and after degradation performance (**B**).

**Table 1 materials-14-07741-t001:** The textural properties of the different catalysts.

Samples	Surface Area (m^2^ g^−1^)	Pore Volume (cm^3^ g^−1^)	Average Pore Size (nm)
NH_2_-MIL-125(Ti)	970.2	0.94	7.33
15% Ni/Ti−MOFs	889.6	0.81	9.03
15% Co/Ti−MOFs	640.7	0.47	4.92
15% Fe/Ti−MOFs	554.9	0.52	7.63

**Table 2 materials-14-07741-t002:** Band gap energy and kinetic parameters for the degradation RhB over different catalysts.

Sample	Band Gap (eV)	R^2^	k_1_ (10^−3^·min^−1^)
NH_2_-MIL-125(Ti)	2.56	0.9846	5.2
15% Ni/Ti-MOFs	2.53	0.9869	5.35
15% Co/Ti-MOFs	2.52	0.9936	5.41
15% Fe/Ti-MOFs	2.40	0.9792	13.49

## Data Availability

All the data is available within the manuscript.

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
