# Peer review of "Enhanced Degradation of Rhodamine B by Metallic Organic Frameworks Based on NH_2_-MIL-125(Ti) under Visible Light"

_materials, 2021, doi:10.3390/ma14247741_

Round 1
Reviewer 1 Report
This work describes the preparation of bare NH2-MIL-125(Ti) MOF particles and MOF particles including different trapped metal ions such as Ni(II),Co(II) and Fe(III). The materials were characterized by different techniques and then assessed as photocatalysts under different conditions using Rhodamine B as a model. Overall I find it an interesting study but more experimental work is needed in some parts to support the conclusions so I recommend reconsideration for publication after addressing the following corrections and questions:
-English language and grammar should be thoroughly checked and corrected. Some phrases are hard to understand and there are typos (e.g. line 22 (bisides), line 77 (metanol)...). Also some scientific terminology is incorrect (e.g. “symmetric lengthen vibrations” instead of symmetric stretching, “vigorous band”??...).
-Some more experimental details should be included. As an example, I consider activation conditions for N2 measurements are important given the fact that MOFs are so sensitive to them.
-I am missing some analysis about the polydispersity degree of the particles (i.e. a statistical deviation could be provided together with the average size). Also regarding polydispersity and morphology, do the authors think that more homogeneous particles could improve the photocatalytic activity? Did they actually try to improve these features in their synthesis conditions?
-Figure 2B: The yellow color is not visible at all. Authors should use a different color.
-Page 8, lines 200-203: Authors should rephrase their statement. A significantly higher removal efficiency was only achieved with 15% Fe on Ti-MOFs, while the inclusion of Ni and Co in the MOF only had a slight improvement.
-Figures 6 and 8: Some of the plots are showing an increase on C/Co during the dark phase. I am missing an explanation for this phenomenon. Also, I suggest changing the way dark/radiation phases are shown (e.g. showing the text “dark” above the arrow, and “light on” below).
-Figure 9: B) is overlapping the graph in A), and a shadowed background and a frame are observed in the graph. The format should be the same as in the other graphs
-Page 11, line 293: “The removal of RhB decreased from 726.88% to 71.19%”, authors should correct the first value.
-I am missing some studies or discussion on the stability of the MOF particles in water, regarding the reusability of the MOF during different cycles. Particularly, I find 3 successive runs (3x120 min= 6 hours) are not enough to state that “the catalyst can be used repeatedly with no discernible variation in photocatalytic activity”. Authors should study the degradation performance and MOF stability for a higher number of runs.
Author Response
Dear Editor and Reviewers,
Please see the attachment

Reviewer 2 Report
This work prepared NH2-MIL-125 MOFs and doped them with different metals such as Ni, Co and Fe. The MOFs were characterized by various techniques such as XRD, FTIR, SEM, BET and UV-Vis and were developed as photocatalysts for RhB degradation. The paper can be accepted for publication only after the following comments are fully addressed.
- In the FTIR spectra, the peaks around 3700 cm-1 are often correlated to -OH from the defects of the MOF. It seems that the peak intensity for Co and Fe doped MOFs decreased compared to the original MOF. The authors need to provide a reasonable explanation for this observation.
- For the Adsorption–desorption isotherms (Fig. 4a), the values of volume absorbed should be shown on the y axis. It seems that for Fe/Ti MOF, hysteresis loop is observed at P/P0=0.6, indicating the presence of mesoporosity. Why did the addition of Fe lead to the mesopores in the MOF?
- In table 1, the pore size of pristine NH2−MIL−125(Ti) is significantly higher than those reported in literature. The large pore size indicates the poor porosity of the MOF, indicating that the synthesis of the MOF was not successful. Moreover, the authors should narrow down the range of x axis of Fig. 4b to 0 to 2.0 nm.
- General speaking, the reviewer has trouble in understanding the concentration reported in the manuscript, such as 2.10-5 M.
- Some parts of the manuscript were not written in standard English, the authors are recommend to ask a professional editor for helps.
Author Response
Dear Editor and reviewers,
Please see the attachment

Reviewer 3 Report
In this paper entitled “Enhanced degradation of Rhodamine B by metallic organic frameworks based on NH2-MIL-125(Ti) under visible light”, Nguyen Thi Kim Oanh and co-workers reported the development of three new bimetallic metal organic frameworks based on the NH2-MIL-125(Ti) structure, but by partially replacing Ti with other metals M (M = Ni, Co, Fe) with a ratio M/Ti = 0.15: 15% Ni/Ti−MOFs, 15% Co/Ti−MOFs, 15% Fe/Ti−MOFs.
After a detailed characterization of the three materials by X-ray diffraction (XRD) analysis, FT-IR spectroscopy, Raman spectroscopy, SEM microscopy, N2 adsorption-desorption measurements and UV-Vis diffuse reflectance spectroscopy, they were tested as photocatalyst to remove Rhodamine B (RhB), as representative example of organic dye, under visible light. The 15% Fe/Ti−MOFs catalyst, which was found to have the highest activity in RhB degradation, was also tested in recyclability tests. In general, the present paper seems interesting: it is the result of a considerable experimental work, and the topic matches with the scope of the journal. Therefore, I believe that it could be worth to be published on MDPI Materials journal, although there are some important issues that need to be carefully addressed before allowing its publication, listed below.
1. A first aspect that in my opinion should clearly transpire in the aim of the work as well as in the conclusions is the novelty brought by this work compared to that previously reported in the literature. Why, in terms of catalytic performance as well as potential technological applicability, should the system developed in this work be better?
2. At the beginning of the introduction, authors focused their attention on the presence of organic matter, and in particular organic colorants, in water streams. This is certainly correct; however, in addition to textile industry, hair dye (DOI: 10.1016/j.toxlet.2007.05.108), leather and paper industries (DOI: 10.1016/S0376-7388(00)00399-9), and luminescent solar concentrator (LSC) technologies (DOI: 10.1002/slct.201800126) also use a large amount of dyes potentially associated with water pollution. I believe that this point should be well specified in the paper, with the support of the above mentioned literature.
3. Another very important aspect that, in my opinion, needs to be definitely correct in the manuscript concerns the wrong use of full stop in the place of middle dot. Indeed, authors used the full stop, in the place of middle dot, in several chemical structures (for example, FeCl3.6H2O, NiCl2.6H2O, CoCl2.6H2O, instead of FeCl3·6 H2O, NiCl2·6 H2O, CoCl2·6 H2O) and especially in numerous mathematical formulas (for example 3.10-5 instead of 3·10-5, 2.10-5 instead of 2·10-5, 4.10-5 instead of 4·10-5, etc.). In this last case, the wrong use of full stop risks to be misleading, since multiplication operations could be easily confused with decimals. Moreover, in other point of the paper, multiplication operations were instead indicated with the letter “x” instead of the usual middle dot (for example in 13.49 x 10-3 min-1, 5.41 x 10-3 min-1, 5.35 x 10-3 min-1 or 5.2 x 10-3 min-1, instead of 13.49 · 10-3 min-1, 5.41 · 10-3 min-1, 5.35 · 10-3 min-1, 5.2 · 10-3 min-1). Please read the document very carefully and replace as suggested above.
4. Another important aspect of the study that in my opinion should be better highlighted and investigated concerns the catalyst recyclability. First, three runs are definitely not enough to guarantee a good recyclability of the catalyst (especially in an industrial application perspective): at least 6-7 runs are mandatory, although in some papers even 20-25 cycles are reported. Second, the removal of RhB decreased from 76.88% (first run) to 71.19% (third run), indicating a clear trend of catalytic activity decreasing ,unlike that is specified by the authors on lines 294-295. I believe that a plausible explanation for this trend should be reported in the paper. Third, please correct “726.88%” with “76.88%” on line 293.
5. I believe the caption of the Figure 7 should be properly corrected, as in the figure were reported UV−Vis absorption spectra of RhB over all the four MOF-based photocatalysts, and not only over NH2−MIL−125(Ti) or 15% Fe/Ti−MOFs.
6. Other minor issue: please carefully check all the numbers used as superscript or subscript in the manuscript (for example in the title, where NH2-MIL-125(Ti) should be changed with NH2-MIL-125(Ti), but also in many other point of the manuscript).
Author Response

(The authors gave the same response as above.)

Round 2
Reviewer 1 Report
I appreaciate the author´s corrections in the manuscript and understand the situation imposed by COVID-19 constraints. I recommend the publication of the paper after the following minor corrections are addressed:
1.-Regarding MOF N2 sorption isotherms, authors should not only include the equipment used (already included) but also the activation conditions (i.e. temperature and time, vacuum/not vacuum).
2.- Scientific terminology: although the english was improved in the manuscript, I am still finding some incorrect terms. Please use the appropriate denomination for the IR absorption peak referenced in “at 1256 cm-1 indexed to the expanded vibrations…” (line 157). The term should correspond to a vibrational mode type, “strongly band” is not a correct term.
Author Response
Dear Editor and Reviewers
Please see the attachment.

Reviewer 2 Report
This reviewer is happy to see that the authors have addressed the issues raised, and therefore recommends its publication in Materials.
Author Response

(The authors gave the same response as above.)

Reviewer 3 Report
In this revised version of the paper “Enhanced degradation of Rhodamine B by metallic organic frameworks based on NH2-MIL-125(Ti) under visible light”, Nguyen Thi Kim Oanh and co-workers addressed quite satisfactorily all the issues listed in my previous report, thus allowing to improve the quality of the work. I read this new version very carefully, in particular all new parts highlighted in yellow, and I thank the authors for their efforts to improve their work. I am convinced that the paper now meets the standards for publication in the MDPI Materials journal, thus suggesting its acceptance in the present form.
Author Response

(The authors gave the same response as above.)
